# Investigations of Working Characteristics of Transferred Arc Plasma Torch Volume Reactor

Žydrūnas Kavaliauskas *, Rolandas Uscila, Romualdas Kėželis, Vitas Valinčius, Viktorija Grigaitienė, Dovilė Gimžauskaitė and Mindaugas Milieška

Lithuanian Energy Institute, Breslaujos Str. 3, LT-44403 Kaunas, Lithuania; rolandas.uscila@lei.lt (R.U.); romualdas.kezelis@lei.lt (R.K.); vitas.valincius@lei.lt (V.V.); viktorija.grigaitiene@lei.lt (V.G.); dovile.gimzauskaite@lei.lt (D.G.); mindaugas.milieska@lei.lt (M.M.)
* Correspondence: zydrunas.kavaliauskaslei@gmail.com

**Abstract:** A transferred arc plasma torch chemical rector was used to process waste formed from mixtures of dry clay powder and hydroquinone. Such reactors are best suited for the treatment of electrically conductive waste. In these types of reactors, the electric arc moves chaotically throughout the entire reactor volume, making it possible to ensure an even temperature distribution in the reaction zones. An analysis of the literature has shown that there are not many study results related to this type of reactor. The novelty of the work is that the behavior of the operating electric arc inside the reactor was recorded by using a high-speed camera. The distribution of the temperature profile at the cooled reactor wall was investigated. The electrical potential difference inside the reactor was also investigated. To better understand the behavioral properties of the electric arc when the reactor is filled with treated material, hydroquinone-contaminated clay was used. In this case, the movement of the electric arc, as well as the probability of its formation, is the greatest at the location where the thinnest layer of the material to be processed is located. In addition, it has been observed that the use of a graphite anode poses problems because, over time, the anode of such a design deforms due to interactions with the electric arc. While analyzing research results, it can be observed that these types of reactors are very suitable for the treatment of electrically conductive materials and for the treatment of small amounts of nonconductive materials when the material occupies a relatively small part of the reactor. A further development of these studies in the future is planned in order to make the reactors as versatile as possible and as suitable as possible for handling the widest range of materials possible.

**Keywords:** plasma; reactor; waste



## 1. Introduction

The problem of waste generated by the development of industrial production and advanced modern technologies is becoming increasingly important. As the consumption of goods and services grows, increasing amounts of secondary raw materials such as various plastics, rubber, medical and chemical waste, etc. are generated [1–3]. Today's science faces new challenges, the solutions of which are related to the sustainable use of waste in the production of various products and equipment, as well as the neutralization and conversion of hazardous waste into environmentally friendly materials. Another important issue that researchers are paying increasing attention to includes energy resources [4–7]. Fossil fuel resources are known to be finite and the use of fossil fuels poses problems such as environmental pollution and the promotion of the greenhouse effect. Therefore, there is an increasing focus on the use of secondary raw materials and waste alternative energy sources, for example, how to safely use waste heat for heating or electricity generation through the incineration, neutralization or other recycling of secondary raw materials [8–10].

There are many developed methods for neutralizing waste of various origins, such as incineration, pyrolysis and so on. However, most of these methods have a significant drawback as the process temperature reaches only 1500 °C, whereas, for hazardous wastes such as furans, chlorine or sulfur compounds and various carbon derivatives, these temperatures are not sufficient for neutralization or recycling processes. In order to neutralize or recycle particularly hazardous and hardly breaking waste, the process temperature must reach at least 1800 °C and above [8–12]. It is not possible to achieve such ambient temperatures using the traditional methods mentioned above. Alternatively, various types of plasma, such as RF plasma, arc discharge formed by plasma torch and other sources can be used to address this problem. Using various types of plasma, the process temperature can be around 5000 °C or even higher. Such temperatures are fully sufficient for the realization of all waste conversion and neutralization processes. However, in order for the waste neutralization process to be fully realized using a plasma environment, it is necessary that waste and other materials remain in the plasma environment for at least 1–2 s. To meet this condition, plasma-chemical reactors of various constructions are designed and developed and the waste particles inside them are retained for the required time and neutralized [13,14].

In practice, volume plasma chemical reactors, non-transferred arc or transferred arc plasma-type reactors, etc., are commonly used. While using plasma neutralization technologies, the end result of waste neutralization is usually a gas. Using plasma technology, the main component of the system is the torch. Torches of various designs, such as AD/DC-type torches, short-term pulsed arc torches, microwave torches, laser-induced plasma torches and others, can be used to obtain thermal plasma [15–17].

The AC arc discharge ensures high energy density and a high temperature range between the two electrodes. At sufficiently high gas flows, the plasma extends in the form of a stream of ionized gas in a sufficiently large volume, which may be limited by the geometric dimensions of the reactor chamber. This design feature ensures that waste particles will remain in the reaction zone for the required time (1–2 s) and will be completely decompose into the gaseous form. Arc plasma generators are divided into nontransferred arc torch and transferred arc torch. Non-transferred arc torch electrodes do not participate in the processing of secondary raw materials and have the sole function of plasma generation. The material to be treated in the transferred arc reactor is placed in a metal grounded vessel that acts as an anode for the system [17–19]. Consequently, the material to be treated must also be relatively electrically conductive or, if impermeable, should not completely cover the entire surface of the anode (certain defined reaction zones) in order to realize an electric arc discharge. This type of plasma device is widely used in metallurgy and in the neutralization processes of electrically conductive waste (e.g., various types of carbon and hazardous metal compounds, etc.). The cooling of this type of plasma device usually takes place using water, and the service life of the electrodes is 200–500 h. The operating capacity of these facilities is usually up to 6 MW. DC-type electric current is mostly often used for these devices to ensure operational stability [19–23].

As it was mentioned, both non-transferred arc torch and transferred arc torch type devices are cooled by water, which makes it possible to use heated water for heating of premises or electricity generation. The main disadvantage of a non-transferred arc torch is that the resulting plasma current is narrow, making it difficult to maintain a constant and sufficiently high enough temperature throughout the volume of the reaction zone. Meanwhile, in a transferred arc torch type plasma chemical reactor, the electric arc moves freely over the entire surface of the anode vessel, heating all its areas evenly. Since in this case the processing of the raw material takes place in the anode region, the temperature of the reaction zone is kept constant throughout the area [1,2].

It has been observed that there is a lack of data in the scientific literature on the performance studies of transferred arc torch type reactors and their use for the neutralization of various wastes. Therefore, the aim of this work is to perform some studies related to plasma arc formation and characteristics in transferred arc torch type devices. Research will

also be carried out to gain a better understanding of the conversion processes of various wastes or other feedstocks in this type of plasma chemical reactor. In addition, the novelty of this work is that the literature usually deals with this type of reactor when processing electrically conductive material. Meanwhile, the authors of this work used an electrically nonconductive material (hydroquinone with clay) to evaluate the plasma flow behavior and operation parameters of the reactor and their mutual influence. One of the tasks of this work was to show the versatility of this type of reactor with respect to the material being processed.

## 2. The Experimental Setup

A transferred arc torch-type volumetric plasma chemical reactor was used for waste neutralization and process research. A prototype of this device was developed at the Plasma Processing Laboratory of the Lithuanian Energy Institute. Both the plasma torch unit and the volumetric reactor walls were cooled with water and air at flow rates of 1.24 kg/s and 6.8 g/s, respectively. The bottom of the plasma chemical rector was made of a conductive graphite layer of material, which also played the role of an anode. The electric arc in the volume reactor was moving chaotically and was formed using air with a flow rate of 0.8 g/s. The recycled material consisting of a mixture of dry clay and hydroquinone (composition of 25% hydroquinone and 75% clay) was dosed onto a conductive graphite surface layer. Hydroquinone was selected as the less active substance. Clay was chosen as the carrier, and hydroquinone was chosen as neutralized waste. With a hydroquinone content of more than 25%, there are technical problems associated with dosing of the mixture into the reactor chamber as the mixture becomes sticky. The distance between the plasma torc and the top layer of conductive material can be varied in the range of 10 cm. The transferred arc torch power is up to 40 kW, and the discharge current is 160 A. Based on the literature, the plasma flow temperature at central part is about 13,000 °C [5]. A schematic diagram of the transferred arc torch-type plasma chemical reactor is shown in Figure 1.

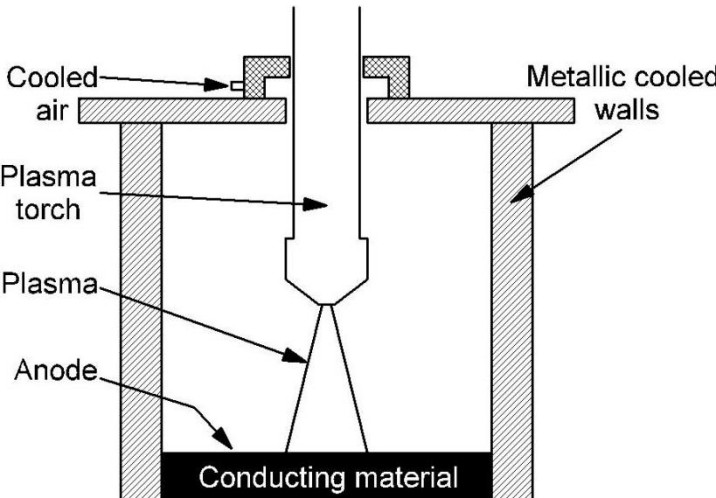

**Figure 1.** The schematic diagram of a transferred arc torch-type plasma chemical reactor.

The investigation of the formation of multiphase flow parameters has been performed also experimentally. A high-speed RedLake MotionPro video camera was used for instantaneous imaging of the plasma arc. A fast, 12-bit CMOS camera (MotionPro from Redlake) equipped with a zoom lens and a neutral density filter was used to visualize the plasma. The camera exposure time is 2–43 μs.

A platinum-rhodium thermocouple was used to record the distributions of plasma flow temperatures and plasma current potential. The Agilent 34972A digital multimeter, which has a data transfer interface with a personal computer, was used to gather thermocouple data.

During the plasma chemical process, when the materials were recycled inside the volume reactor, a gas analyzer SWG 300-1 was used to analyze the composition of the emitted gas, to which the gas was fed using a water-cooled probe. The probe is structurally made of a double metal wall, making it possible to cool the entire probe (the entire surface of the probe). The morphology of the materials used for neutralization was evaluated using an electronic scanning microscope Hitachi S-3400N with magnification ranging from 20 to 100,000 times and 5 nm resolution.

## 3. The Results of the Experiment

Figure 2 shows plasma flow images obtained using a high-speed RedLake MotionPro video camera. By analyzing the images, the electric arc of the ionized gas can be observed to move chaotically inside the volume plasma chemical reactor. An electric arc discharge is generated between the conductive bottom of the reactor, which acts as an anode and the plasma torch. The primary discharge of the electric arc is obtained using a high-voltage 20 kV oscillator. The resulting strong electric field ionizes the plasma-forming gas. After the initial discharge, the electric arc enters the self-discharge phase and is supported by the impact ionization process. Impact ionization requires an additional electron source for which its role is played by the hafnium built into the cathode. The electrons are emitted when a thermoelectric emission phenomenon occurs. Figure 2a–c show the discharge of an electric arc when no material is treated inside the plasma chemical reactor, and the bottom is made of a conductive graphite acting as an anode. By analyzing these images, it was observed that the electric arc moves over the entire surface of the graphite anode. The chaotic movement can be explained by the fact that the arc is formed at the location of the volume reactor where the conditions for impact ionization are mostly favorable. An electric arc occurs where the lowest energy consumption is required to form it. Figure 2d–f show the electric arc discharge when the clay material is mixed with hydroquinone on the graphitic bottom of the volumetric reactor. This material was selected for reactor operation studies and the visual analysis of plasma arc behavior. As we can see in this case, the electric arc moves chaotically. The particles of the treated material are fed by a special dispenser onto the surface of the anode, which is covered with a layer of uneven thickness. Since clay and hydroquinone are poor conductors, the probability of electric arc formation is the greatest at the point where the particle layer is the thinnest. In addition, the movement of the arc is affected by the fact that the particles of the material to be processed melt on the surface of the anode; thus, the thickness of the layer is constantly changing while simultaneously influencing the energy characteristics of the electric arc's formation. In transferred arc torch-type plasma type reactors, there is one constructive problem that the thickness of the anode bottom changes due to the constant interaction with the electric arc. As a result, the service life of the plasma chemical reactor decreases and the energy characteristics of the electric arc formation varies.

The thermal characteristics of the plasma chemical reactor were evaluated during the study. A special opening was made in the wall of the plasma chemical reactor through which the temperature distribution profile at the cooled wall of the plasma chemical reactor was measured using the platinum–rhodium thermocouple. The points at which the temperature was measured are shown schematically in Figure 3. The temperature measurement step was 5 mm from one edge of the opening, which is considered as the zero reference point to a distance of 60 mm near the other edge. The very center of the opening is at 30 mm (this is clearly seen in Figure 3).

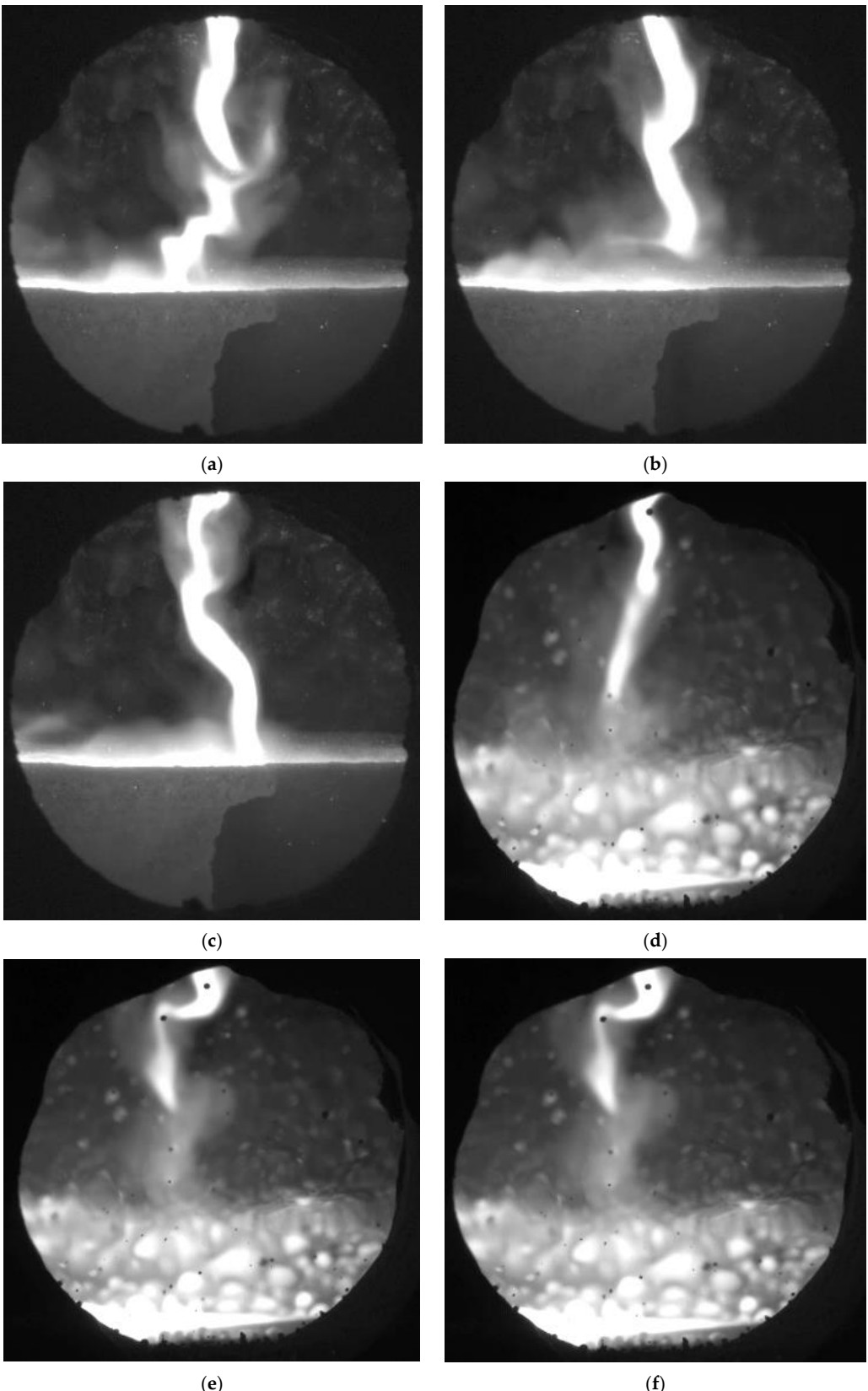

(**a**)　　　　　　　　　　　　　　　(**b**)

(**c**)　　　　　　　　　　　　　　　(**d**)

(**e**)　　　　　　　　　　　　　　　(**f**)

**Figure 2.** High speed video camera images with a frame exposure time of 1 µs and frames per second rate of 500 fps: when (**a–c**), there is no material to be processed in the reactor and (**d–f**) is when the reactor is filled with the clay to be treated.

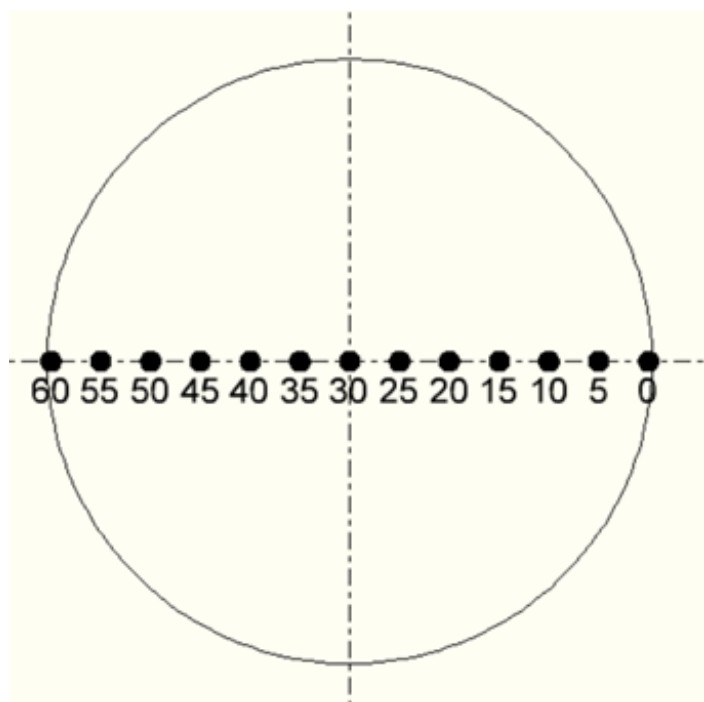

**Figure 3.** The dimensions of the opening for the temperature profile measurements at the cooled wall and the steps of the thermocouple movement. The distance between the anode surface and the temperature measurements plane is 100 mm.

Figure 4 shows the measurement results of the temperature profile. The measurements were made using the special hole made in the wall of the reactor mentioned above for measurements. During the temperature measurement, the material was treated inside the reactor. An automatic positioning device was used to change the position of the platinum–rhodium thermocouple nozzle. As it can be seen from the results presented, the lowest temperature value of 640 °C was obtained when the thermocouple is in the starting 0 mm position. The temperature rises up to 1100 °C when measured at a position of 30 mm corresponding to the center of the measuring hole. Moving further towards the 60 mm position, the temperature reaches 800 °C. As it is observed to be moving along the entire length of the orifice diameter, the temperature distribution is uneven [1–3]. The highest temperature value is at the middle of the opening and the lowest values are at the edges of the opening. The uneven distribution of temperature values can be qualitatively explained by gas flow properties and the distribution of the degree of ionization. The degree of ionization of the plasma-forming gas at the center of the orifice is the highest, which results in the highest concentration of plasma-forming electrical particles at this location, which is why we have the highest temperature at this location. Meanwhile, at the edges of the orifice, the degree of ionization of the plasma-forming gas is significantly lower; thus, the concentrations and temperature of free electrical particles are lower. This temperature profile is also determined by the phenomena of gas ejection, which makes both the density and velocity of the gas uneven.

The distribution of the plasma flow potential difference along the volume reactor was measured during the experiment. A metal probe was used for these measurements when one end of the cable connected to the probe and the other to a grounded electrode. The scheme used to measure the electric arc potential difference is shown in Figure 5. The measurements were performed at five points in every 50 mm.

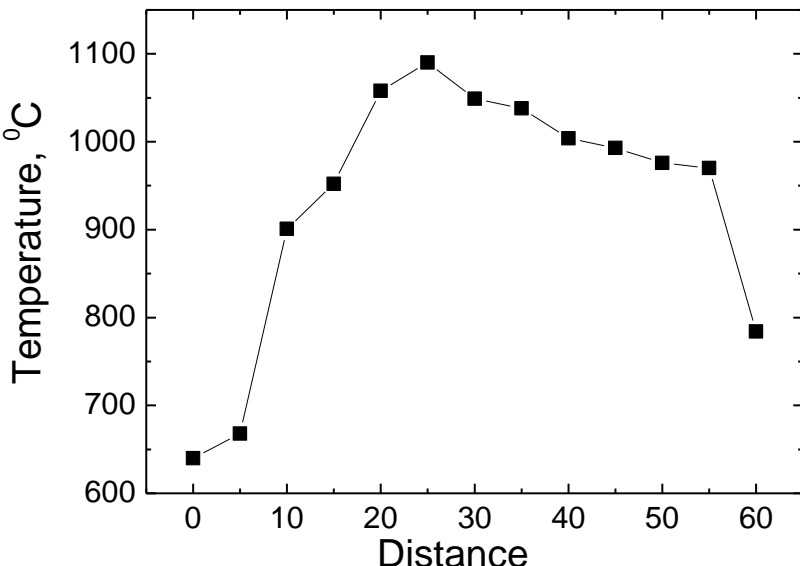

**Figure 4.** The dependence of temperature profile on the diameter of the cooling wall hole.

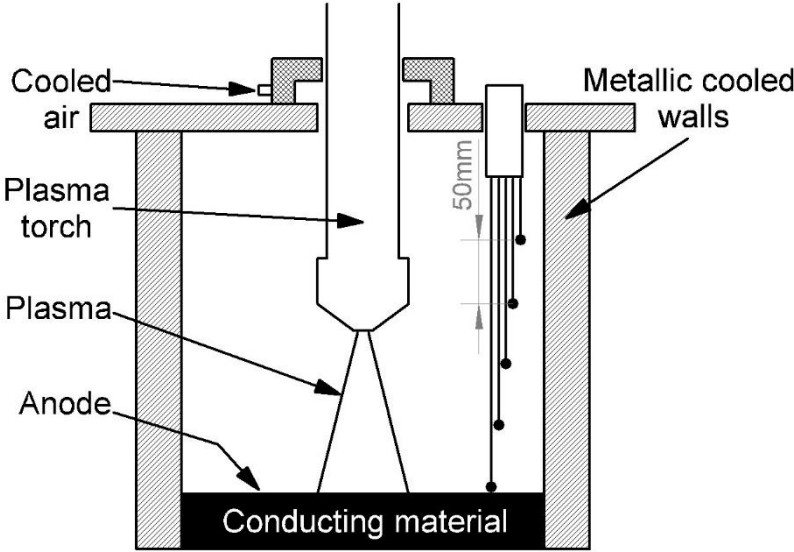

**Figure 5.** The schematic diagram for measuring the difference in plasma electric arc potentials. Measurements were performed with a cooling probe every 50 mm.

The values obtained for the potential difference during the experiment are average because the motion of the electric arc is chaotic throughout the reactor volume. The plasma arc potential difference measurements were performed at certain time intervals inside the reactor in the presence of the treated material. The results of the measurements are provided in Table 1. As observed, the difference in the potentials along the length of the reactor changes slightly at some point in time. At 6 min of the experiment, the potential difference in the reactor volume varies between 2.2 and 3.2 V. This result can be explained by the fact that, at the beginning of the experiment, the treated material (clay and hydroquinone mixture) did not melt and covered most of the surface of the graphite anode. Meanwhile, the inequality of the potential at different points is also determined by the fact that the density of the plasma-forming gas is not equal throughout the volume of the reactor due to the unequal temperature inside the reactor. In addition, the unequal distribution of electrical potential was determined due to the fact that the plasma arc moves chaotically throughout the reactor. When the duration of the experiment reaches 40 min, the potential of the electric arc increases significantly and its values change in the range of approximately

40–43 V. At this time value, the processed clay transitions from the solid phase to the liquid electrically conductive phase. These phase transitions of the material also affect potential difference values. In the case of conductive liquid phase, the electrical potential difference increases significantly. The formation of an arc in a thinner location of a nonconductive coating is associated with the formation of a negative surface charge, which prevents electrons from leaving towards the anode. On a thinner layer, this potential is lower; therefore, the conditions for an arc formation are more favorable here. The values of the electrical potential difference are also influenced by the changing distance between the plasma torch and the surface of the treated material, as part of the liquid phase material is removed and, in addition, the relief of the liquid surface is constantly changing [24]. The change in potential difference is also influenced by the fact that evaporation takes place during the melting of the processed material, which changes the composition and density of the gas inside the reactor. As the treated material evaporates, the electric arc becomes increasingly diffuse (more scattered than when there is no material to be treated in the reactor), which affects the dynamics of the potential difference. Looking at all these reasons for the change of electrical potential, it is difficult to quantify their impact.

**Table 1.** Plasma in a chemical reactor and the electric potential of an electric arc at different locations in the reactor.

| | Electric Arc Potential Difference | | | | |
|---|---|---|---|---|---|
| Time, min. | 1 | 2 | 3 | 4 | 5 |
| 6 | 2,9 | 3,12 | 3,2 | 2,4 | 2,2 |
| 12 | 12,6 | 13,5 | 13,8 | 13,9 | 13,8 |
| 17 | 30 | 30,6 | 30,1 | 27,7 | 28,5 |
| 25 | 54 | 50 | 50,1 | 51,3 | 52,1 |
| 35 | 63 | 69 | 68 | 63 | 48 |
| 40 | 43 | 42 | 42 | 41 | 40 |

A scanning electron microscope was used to evaluate the external structure of the primary raw clay/hydroquinone material and the waste generated after the plasma chemical process in the plasma chemical reactor. The obtained SEM imaging results are shown in Figure 6. Figure 6a is a view of the primary feedstock before entering the reactor. As observed, the structure of the primary raw material consists of various irregularly shaped microformations. While analyzing this SEM image, it was observed that the geometric dimensions of irregularly shaped structures are very different and vary in the range of 1–10 μm. Melting, boiling and evaporation of the feedstock take place inside the reactor during the plasma chemical process (these processes are the main ones that change the surface relief of the processed material). Therefore, it is natural that the amount of primary raw material is significantly reduced by melting or vaporization. Figure 6b shows an image of a material that is already processed during the plasma chemical process. The surface structure comprises various irregularly shaped microstructures. In this case, due to the effect of the process conditions, the microformations have smaller geometrical dimensions than the primary raw materials. The geometric dimensions of various microstructures vary approximately in the range of 0.5–5 μm.

In order to neutralize certain harmful substances (e.g., hydroquinone), it is necessary to use a secondary carrier so that the substance to be neutralized can be easily injected into the plasma chemical reactor. In this case, a dry clay powder mixed with hydroquinone was used as the carrier. The EDS method was used to estimate the chemical composition of this primary raw material. The results of the EDS studies are presented in Figure 7. As the results of the estimation show, the primary raw material comprises silicon 13.55%, calcium 6%, aluminum 5.7%, carbon 9.16%, iron 1.45%, potassium 1.77% and magnesium 1.34%. The analysis of the results shows that the material contains about 62% oxygen. Part of the free oxygen is concentrated in surface micropores. The rest of the oxygen is accumulated in the form of oxides, as part of the processed raw material does not comprise pure metals but

comprise metal oxides instead. Some of the oxygen present at high temperatures can act as a catalytic gas, forming volatile compounds and accelerating the decomposition reactions of the material being treated. Sodium, tungsten, titanium and lead were also observed in the analysis of EDS studies, but their content is generally insignificant and ranges only in the range of about 0.01–0.2%. The results of the EDS studies were analyzed according to the atomic mass of the elements.

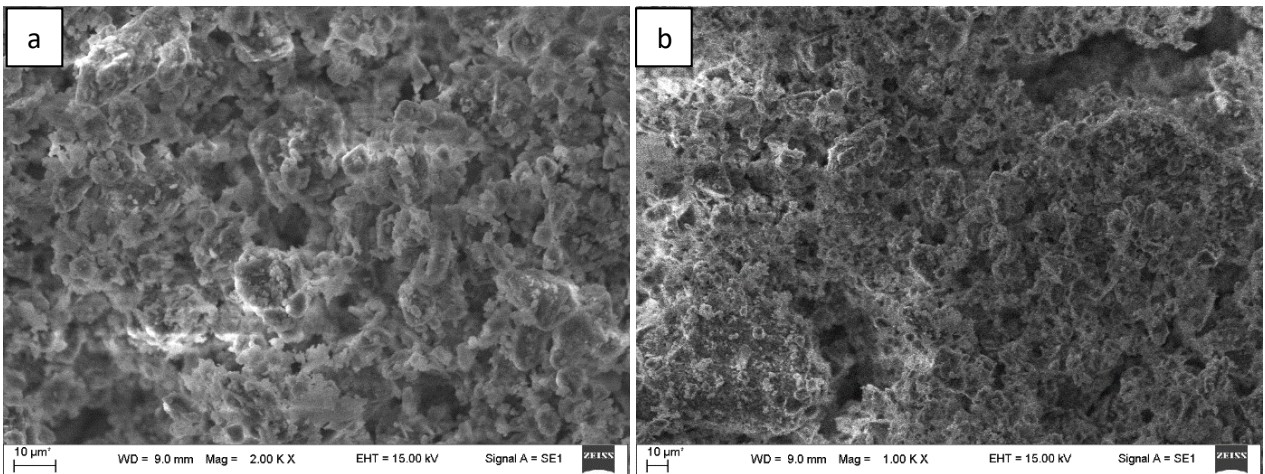

**Figure 6.** The SEM images of clay powder mixed with hydroquinone and intended for reactor treatment: (**a**) feedstock before entering the reactor and (**b**) feedstock after plasma chemical treatment.

| El | AN | Series | Net | unn. C [wt.%] | norm. C [wt.%] | Atom. C [at.%] | Error (1 Sigma) [wt.%] |
|----|----|--------|-----|------|------|------|---------|
| O  | 8  | K-series | 12686 | 47.37 | 48.51 | 61.91 | 6.44 |
| Si | 14 | K-series | 19544 | 18.20 | 18.63 | 13.55 | 0.80 |
| Ca | 20 | K-series | 6405 | 11.49 | 11.77 | 5.99 | 0.41 |
| Al | 13 | K-series | 5874 | 5.58 | 5.72 | 4.33 | 0.30 |
| C  | 6  | K-series | 630 | 5.26 | 5.39 | 9.16 | 1.39 |
| Fe | 26 | K-series | 1203 | 3.88 | 3.97 | 1.45 | 0.20 |
| K  | 19 | K-series | 2372 | 3.31 | 3.39 | 1.77 | 0.15 |
| Mg | 12 | K-series | 1442 | 1.56 | 1.60 | 1.34 | 0.13 |
| Ti | 22 | K-series | 206 | 0.47 | 0.48 | 0.20 | 0.06 |
| Na | 11 | K-series | 194 | 0.29 | 0.30 | 0.27 | 0.06 |
| Pb | 82 | M-series | 71 | 0.15 | 0.16 | 0.02 | 0.05 |
| W  | 74 | L-series | 5 | 0.08 | 0.08 | 0.01 | 0.06 |
| | | Total: | | 97.64 | 100.00 | 100.00 | |

**Figure 7.** The EDS measurements of primary feedstock before entering the reactor.

As gases of various compositions are formed during processing the primary feedstock inside the plasma chemical reactor, a gas analyzer was used to evaluate their composition. The gas was collected from inside of the reactor using a specially cooled probe and fed to separate cells in the analyzer for gas analysis. When measuring the gas composition,

the duration of the experiment was 16 min. The dynamics of oxygen and carbon dioxide content during the experiment is shown in Figure 8. As observed, the oxygen content at the initial time is about 20%, which corresponds to the total oxygen content in the atmosphere. During the plasma chemical treatment of the primary raw material and at the sixth minute of the experiment, the total $O_2$ content was reduced by about 5%. This trend lasts for up to 12 min of the experiment, with the total $O_2$ content inside the reactor reaching about 17%. This decrease is associated with dissociation and the fact that oxygen combines with nitrogen gas. Part of the nitrogen and oxygen also reacts with the hot walls of the plasma reactor. As a result, oxygen concentration decreases. The total $CO_2$ content at the initial point of the plasma chemical process is equal to 0%. An insignificant increase in these gases was observed at 4 min of the experiment. As it can be observed from the measurement results, the $CO_2$ inside the plasma chemical reactor increased linearly until the end of the experiment at 16 min. This can be explained by the increase in the amount of atomic carbon as plasma flow temperature increases, as more and more organic products are processed at higher temperatures. During this time, $CO_2$ content increases up to about 5%. An increase in the concentration of these gases is also a reason for the decrease in total oxygen [21].

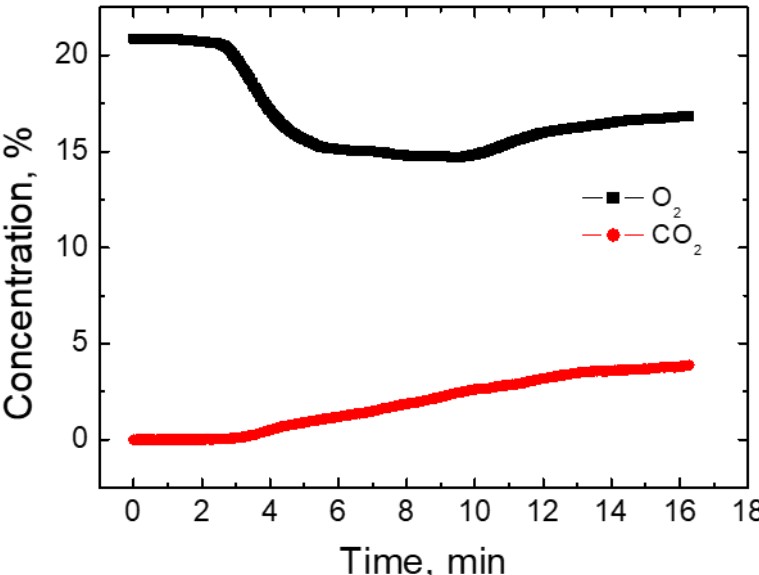

**Figure 8.** The dependence of gas concentration on the duration of the experiment.

The total percentage of $SO_2$, $NO_2$ and $NO$ inside the reactor was also measured during the treatment of the clay contaminated with hydroquinone (Figure 9). When analyzing measurement results, the amount of these gases formed in the total volume was not large. The largest amount of NO gas was formed, and the increase was observed at the second minute of the experiment. NO content increased until the 10th minute of the experiment and reached about 0.027%. Meanwhile, during a further experiment, the total amount of NO remained approximately constant. Gas composition measurements showed a small amount of $NO_2$ during the experiment. A steady increase in $NO_2$ was observed throughout the experiment and reached about 0.0025% at the end of the experiment at the 16th minute.

Sulfur dioxide concentrations were also measured during the experiment. As the analysis of the experimental results shows, the dynamics of $SO_2$ change is very similar to the change of $NO_2$ concentration [21]. A slight increase in $SO_2$ was observed throughout the experiment. The highest $SO_2$ concentration was obtained at the 16th minute of the experiment and reached about 0.001%. The formation of hydrogen gas and $C_xH_x$ compounds was also monitored during the experiment, but no concentration of $H_2$ gas was observed during the entire experiment. The same result (concentration in the volume reactor 0%) was observed while measuring the concentrations of $C_xH_x$ compounds.

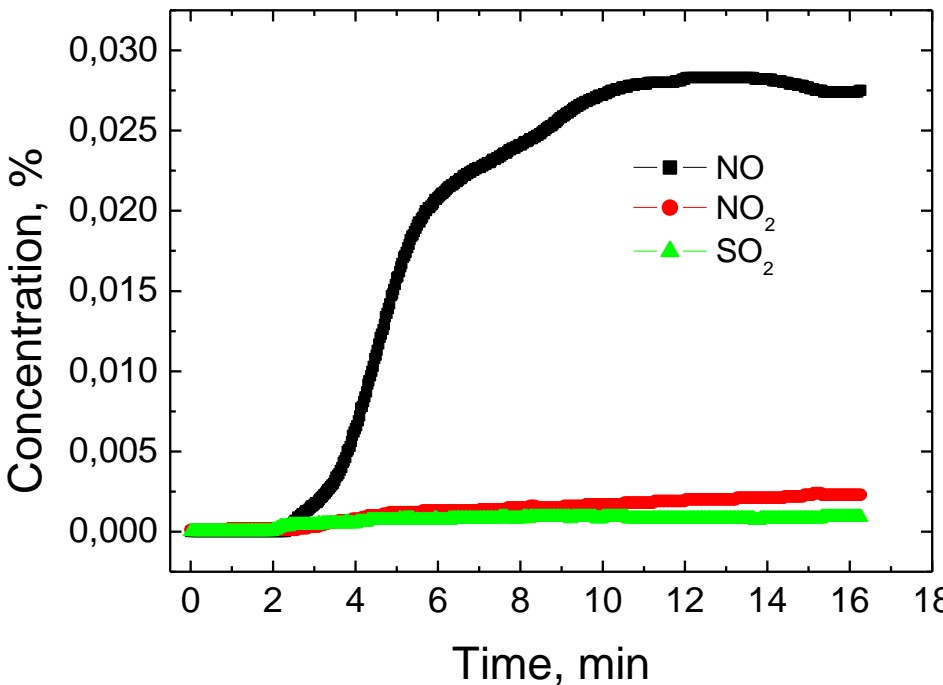

**Figure 9.** The dependence of gas concentration throughout the experiment.

### 4. Conclusions

The movement of the electric arc, as well as the probability of its formation, is the greatest at the location where the thinnest layer of the material to be processed is located. Although the graphite anode portion of the reactor deforms slightly over time (over a several hundred operating hours) due to the chemical interaction between the plasma-forming gas, the material being processed and the graphite anode surface, this is not a major problem because graphite is a relatively inexpensive material. In addition, the graphite anode in the plasma chemical system is a rapidly replaceable part, which does not cause major problems in the operation of this type of reactor. As the measurements show due to the flow characteristics of the plasma-forming gas flow and the unequal distribution of the degree of ionization in the plasma flow, the maximum temperature obtained inside the orifice reaches 1100 °C. When measuring the electric potential difference of the electric arc, it was observed that due to the change of the physical state and amount of the processed material, the potential increases from 2.2 to 43 V when the experiment lasted for 40 min. By analyzing the SEM images, it was observed that the processed raw material is composed of irregularly shaped formations. Meanwhile, the material remaining after processing comprised smaller microproducts than compared to the primary raw material before entering the reactor. When studying the composition of the formed gas by the gas analyzer, it was observed that molecular oxygen was reduced by about 3–5%, while $CO_2$ content increased and comprised 5% of the total concentration of the emitted gas. During the experiment, NO concentration was evaluated, the increase of which occurred up to 10th minute of the experiment and reached about 0.027%. At that time, when assessing the dynamics of $SO_2$ and $NO_2$ change, a slight and steady increase in the amount during the entire experiment was observed. This type of reactor is favorable because the chaotic movement of the electric arc throughout the reactor volume ensures an even temperature distribution in the reaction zones. Studies show that this type of reactor can be used not only for the treatment of electrically conductive materials but they also can be successfully applied for the treatment of nonconductive materials.

**Author Contributions:** Conceptualization, Ž.K. and R.U.; methodology, R.K.; software, V.V.; validation, V.G., D.G. and M.M.; formal analysis, Ž.K.; investigation, Ž.K., R.U.; resources, V.V.; data curation, D.G.; writing—original draft preparation, Ž.K., R.U.; writing—review and editing, Ž.K.; visualization, V.V.; supervision, V.V.; project administration, V.V.; funding acquisition, V.V. All authors have read and agreed to the published version of the manuscript.

**Funding:** This research has received funding from European Regional Development Fund (project No. 01.2.2-LMT-K-718-01-0069) under grant agreement with the Research Council of Lithuania (LMTLT).

**Institutional Review Board Statement:** Not applicable.

**Informed Consent Statement:** Not applicable.

**Data Availability Statement:** Not applicable.

**Conflicts of Interest:** The authors declare no conflict of interest.

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
