# Peer review of "Investigations of Working Characteristics of Transferred Arc Plasma Torch Volume Reactor"

_applsci, doi:10.3390/app12052624_

Round 1
Reviewer 1 Report
This is an experimental work describing the characterization and analysis of a transferred arc plasma torch chemical rector intended to process electrically conductive waste. The waste was engineered by mixing dry clay powder with hydroquinone (an important industrial chemical, which is harmful to living organisms, a potent environmental pollutant and a carcinogen). The motion pattern of the electric arc within the reactor was investigated using a high-speed image capturing camera. Spatial distribution of temperature in the reactor chamber was determined by thermocouples, and the distribution of the electric potential by metal probes. The harmful waste was characterized by SEM (morphology) and by EDS (composition) before and after plasma treatment. The composition of the exhaust gases which were the main product of waste neutralization process was characterized by a gas analyzer. One of the goals of the presented research was to fulfill the gaps in the practical knowledge on the performance of transferred arc torch type reactors and their use for the neutralization of various types of wastes.
The science in the manuscript is sound and solid, without incorrect assumptions and errors. The methodology applied is appropriate and the disclosed data are sufficient to enable other teams to repeat the procedure. The manuscript is well written, with an exemplary style and structuring, and English is very good throughout. The only problem I encountered is that some descriptions could have been shortened without any detriment of the overall text. The obtained experimental results are relevant and important from the point of view of environmental remediation and harmful waste neutralization. The references are appropriate and well chosen. I believe that the manuscript should be accepted for publication, provided that corrections are made. A point-by-point list of my main concerns is given below.
- Please describe clearly and concisely in Abstract and Introduction what is the main novelty of your manuscript.
- A short description of the main conclusions and the main planned fields of future use are also missing from Abstract. An abstract must be a self-contained text which represents a paper in little and contains all the necessary information in a self-contained form.
- Why only hydroquinone as a pollutant was considered? The gaps in the knowledge on plasma-assisted neutralization are much wider and are related with numerous other pollutants. Why at least some other types of organic waste were not analyzed? Do mention this in the revised version.
- In several places in the manuscript it has been said that the transferred arc torch-type plasma chemical reactors are best suited for the treatment of electrically conductive wastes. Yet the type of the waste analyzed in the manuscript is not electrically conductive, except when melted. Do explain this, please.
- Did you check the homogeneity of discharge distribution throughout the waste mass? You wrote that the temperature distribution was inhomogeneous. Was the entire amount of the feedstock neutralized?
- How was the inside temperature of 13000 Celsius measured (line 114)? No temperature probes made of solid material would survive it.
- There is a slight technical problem with Fig 2: in the pdf version, the rectangular shapes containing the notations a, b, c, d e, f are scattered all around, and most of them appear not to be in their proper positions. One of them (b) even went so far up to cover the middle of the preceding paragraph. To avoid it, in my experience, it is the most convenient when all figure are fully converted to bitmaps (including their notations and insets) with appropriate resolution before inserting them into the manuscript. In that way one is always sure that everything will be properly positioned within each figure.
- Please denote the probes for potential distribution measurement at the left part of Fig. 5.
- You wrote, “dry clay powder mixed with hydroquinone was used as the carrier” (line 254). Isn’t the dry clay powder the only carrier here, while hydroquinone is the harmful pollutant to be eliminated?
- Conclusion section could be shortened by deleting needless repetitions of what was already said before and by compacting the overly detailed procedure/measurements description. It should concentrate instead to the main conclusions and to the outlook of the future work.
Reviewer 2 Report
The manuscript investigates the plasma arc formation between transferred arc torch and treated materials in a chemical rector for the neutralization of waste production. Though the work reports some experimental results and makes analysis on the results qualitatively, the underlying physics is not very clear and lacks of evidences. I think this manuscript reads more like an experiment report and is full of unclear explanations. It should be re-written according to a scientific paper style, not just a summary of an experiment. Meanwhile, the novelty of the manuscript is also not clearly described. There are also many misprints over the manuscript. Hope the author revise the manuscript from the introduction and give clear and reasonable explanations to the results.
As a help to improve the article, I will give a number of possible tips.
- The introduction raises some highly ambitious expectations with respect to investigation of the plasma arc formation and the conversion processes of waste production. But the only clear conclusion is that an arc moves chaotically and it impacts the treated material followed by oxygen evaporation and CO2, NO, …. formation. These effects are well known and not new. Authors should focus on improving the performance or explaining the physical mechanisms in chemical reactor.
- It is commonly used by PSI community during many years to describe a plasma in term of temperature of charged particles (electrons and positive ions), plasma density, space and floating potentials. There are lots of standard methods to measure them. From this point of view, it is not clear “The plasma flow temperature inside the volume plasma chemical reactor reached about 13000 0 C.” Measurements with a thermocouple will give the surface temperature of the thermocouple itself, but not of “distributions of plasma flow temperatures”.
- Using insulated thermocouple it is possible to measure a floating plasma potential distribution. What is “Electric arc potential difference”? As for my knowledge, the potential of insulated electrode is determined by electrons and ions flow from plasma, and should have negative sign relative to plasma (space potential) due to higher electrons velocity. Author should carefully consider the potential distributions based on this fact.
- What is “water-cooled probe” for gas injection? Maybe water-cooled nozzle?
- The formation of an arc in a thinner place of a non-conductive coating, in my opinion, is associated with the formation of a negative surface charge, which prevents the electrons from leaving for the anode. On a thinner layer, this potential is less, therefore, the conditions for an arc bundle are more favorable here.
- In the case of evaporation of the substrate material, it will be ionized by plasma electrons, with the formation of a bipolar ion current. See, for example https://doi.org/10.1016/j.vacuum.2021.110142
- Figure 3, in my opinion, is excess.
- Figure 5 should be combined with Figure 1.
- There is need more detailed discussion over Figure 6, because it is not clear why they are here at all. What the physics behind them?
In a word, I don't recommend to publish the paper in this form.
Reviewer 3 Report
Review
(Manuscript ID: applsci-1604048)
Destruction of wastes of different nature is an important task. A special role in its solution is played by the use of low-temperature plasma. Transferred arcs are often used to produce plasma, and a number of technological devices have already been developed. However, information about the discharge is not enough and, moreover, each device requires a detailed study to optimize its technological characteristics. One of such devices is explored in the peer-reviewed article.
The authors obtained many results, but after reading the article, a number of questions arose, the answers to which are not in the article.
Comments:
The most important comment is related to the purpose of the work: The authors write that the design of experimental setup used was unsuccessful (page 11, lines 320-322: “In addition, it has been observed that the use of a graphite anode presents a problem because over time the anode of such a structure deforms due to its interaction with the electric arc.”). If it so, does it make sense to study the working characteristics of an unsuccessful design of a plasma-chemical device?
Other comments:
- Why was dry clay and hydroquinone (composition of 25% hydroquinone and 75% clay) chosen as the material to be processed?
- What was the discharge current?
- Page 3, line 114: «temperature inside the volume plasma chemical reactor reached about 13000 0». Is it really true? The text does not mention this temperature.
- What was the temperature of the cooled walls of the reactor? Is it was close to the temperature in positions marked as “0” and “60” in fig. 4?
- Caption to fig. 2 should be changed to write the frame exposure time and time interval between frames.
- At what height from the anode is the temperature data obtained? It is desirable to indicate the measurement plane in Fig. 2.
- Fig. 4: It is not clear whether the temperature values in fig. 4 are measured in an empty reactor, or in a reactor with the treated substance? If it was measured in empty reactor, how did the temperature change in the presence of the treated substance?
- “A metal probe was used for these measurements, with one end connected to the outer metal wall of the plasma torch and the other connected to a grounded electrode.” This text is not clear.
- What information about the processes in plasma does the measurement of the potential give and why does its value pass through a maximum at times of 35 s?
All what has been said above can be summarized by the statement that the article cannot be recommended for publication in the presented form, and the article requires major revision.
Round 2
Reviewer 1 Report
The authors responded to my suggestions and concerns and performed appropriate modifications. I believe the manuscript can be published in its present form.
Reviewer 2 Report
The authors carefully considered all the comments and corrected all the slippery points. It is clearly understood the article nowelty and the methods of measurements in present form. My previous negative review was due to the authors bugs in the experiments description, which gave me reason to doubt in the used methods. But the authors proved their professionalism.
In a word, I suggest to accept the article.
Reviewer 3 Report
The authors answered the reviewer's questions and made changes to the text of the article.
I believe that the revised article can be published.